# UK Patient Access to Low-Protein Prescription Foods in Phenylketonuria (PKU): An Uneasy Path

**DOI:** 10.3390/nu17030392

**Published:** 2025-01-22

**Authors:** Sharon Evans, Cameron Arbuckle, Catherine Ashmore, Sarah Bailey, Giana Blaauw, Wahid Chaudhry, Clare Dale, Anne Daly, Breanna Downey, Jane Dundas, Charlotte Ellerton, Suzanne Ford, Lisa Gaff, Joanna Gribben, Anne Grimsley, Melanie Hill, Laura Murphy, Camille Newby, Natalia Oxley, Rachel Pereira, Alex Pinto, Rachel Skeath, Alexa Sparks, Simon Tapley, Allyson Terry, Georgina Wood, Alison Woodall, Katie Yeung, Anita MacDonald

**Affiliations:** 1Birmingham Children’s Hospital, Birmingham B4 6NH, UK; catherine.ashmore@nhs.net (C.A.); a.daly3@nhs.net (A.D.); alex.pinto@nhs.net (A.P.); anita.macdonald@nhs.net (A.M.); 2University College London Hospitals NHS Foundation Trust, London NW1 2BU, UK; cameron.arbuckle@nhs.net (C.A.); c.ellerton@nhs.net (C.E.); 3University Hospital of Wales, Cardiff CF14 4XW, UK; sarah.bailey3@wales.nhs.uk; 4Queens Medical Centre, Nottingham NG7 2UH, UK; giana.blaauw@nhs.net; 5Great Ormond Street Hospital, London WC1N 3JH, UK; wahid.chaudry@gosh.nhs.uk (W.C.); rachel.skeath@gosh.nhs.uk (R.S.); 6University Hospitals Birmingham, Birmingham B15 2GW, UK; clare.dale@uhb.nhs.uk; 7Sheffield Children’s Hospital, Sheffield S10 2TH, UK; breanna.downey1@nhs.net; 8Royal Belfast Hospital for Sick Children, Belfast BT12 6BA, UK; jane.dundas@belfasttrust.hscni.net (J.D.); anne.grimsley@belfasttrust.hscni.net (A.G.); 9Southmead Hospital, North Bristol, Bristol BS10 5NB, UK; suzanne.ford@nbt.nhs.uk; 10Addenbrookes Hospital, Cambridge CB2 0QQ, UK; lisa.gaff@nhs.net; 11Evelina London Children’s Hospital, London SE1 7EH, UK; joanna.eardley@gstt.nhs.uk; 12Northern General Hospital, Sheffield S5 7AU, UK; melanie.hill13@nhs.net; 13Royal Victoria Hospital, Belfast BT12 6BA, UK; lauraa.murphy@belfasttrust.hscni.net; 14Bristol Royal Hospital for Children, Bristol BS2 8BJ, UK; camille.newby@uhbw.nhs.uk; 15St Lukes Hospital, Bradford BD5 0NA, UK; 16Norfolk & Norwich University Hospital, Norwich NR4 7UY, UK; rachel.pereira@nnuh.nhs.uk; 17Newcastle Upon Tyne Hospitals NHS Foundation Trust, Newcastle upon Tyne NE1 4LP, UK; alexa.sparks1@nhs.net; 18Bristol Royal Infirmary, Bristol BS2 8HW, UK; simon.tapley@uhbw.nhs.uk; 19Alder Hey Children’s Hospital, Liverpool L14 5AB, UK; allyson.terry@alderhey.nhs.uk; 20Royal Manchester Children’s Hospital, Manchester M13 9WL, UK; georgina.wood@mft.nhs.uk; 21Salford Royal Hospital, Salford M6 8HD, UK; alison.woodall@nca.nhs.uk; 22Guy’s and St Thomas’ Hospital London, London SE1 7EH, UK; katie.yeung3@nhs.net

**Keywords:** phenylketonuria, PKU, low-protein foods, dietitian

## Abstract

Background: Special low-protein foods are essential in the dietary treatment of phenylketonuria (PKU). In the UK, these are available on prescription through the General Practitioners (GPs) and distributed via nutritional home delivery companies or pharmacies. Methods: A 58-item online non-validated semi-structured questionnaire was emailed to British Inherited Metabolic Disease Group (BIMDG) dietitians and dietetic support workers (DSW)/administrators working in PKU to ascertain the main system issues and errors with the supply of low-protein prescription foods (LPPF). Results: 73% (*n* = 53/73) of dietitians and 72% (*n* = 18/25) of DSW/administrators responded. A total of 80 questionnaires (representing 44 paediatric and 36 adult PKU centres) were completed. A total of 50% (*n* = 40/80) of respondents reported patient/caregiver problems accessing LPPF at least weekly. The most common problems were unavailable products (82%), missing LPPF in deliveries (79%), and delayed deliveries (66%). For 64% of respondents, >25% of their patients had recurring problems accessing LPPF, and 69% of respondents spent ≥1 h/week and 11% >5 h/week correcting LPPF patient supply issues. The most common foods patients experienced supply issues with were bread (96%), pasta/rice (41%) and milk replacements (35%). This was associated with GP prescription errors (65%), LPPF prescriptions sent to incorrect dispensers/suppliers (60%), and manufacturer supply issues (54%). Problems with patients/caregivers included not ordering LPPF in a timely way (81%), not responding to messages from home delivery companies (73%) and poor understanding of the ordering process (70%). The majority (93%) of respondents reported that prescription issues impacted their patients’ blood Phe control. Suggestions for improving access to LPPF included centralisation of the system to one supplier (76%) and apps for ordering LPPF (69%). Conclusions: The supply of LPPF for PKU in the UK is problematic; it may adversely affect the ability of patients to adhere to dietary management, and a review investigating patient access to LPPF is urgently required.

## 1. Introduction

Phenylketonuria (PKU) is a rare inherited disorder of metabolism caused by a deficiency or absence of the hepatic enzyme phenylalanine (Phe) hydroxylase, resulting in the partial or complete inability to catalyse the hydroxylation of the essential amino acid Phe to tyrosine [1]. This leads to accumulation of Phe in the blood and brain, tyrosine deficiency, and neurological impairment. Whilst there are some pharmacological treatments such as pegvaliase (a subcutaneously administered Phe ammonium lyase enzyme), and sapropterin (an orally administered tetrahydrobiopterin dihydrochloride that acts as a chaperone), their use is limited to subgroups of patients. Pegvaliase is aimed at patients aged ≥16 years with a blood Phe ≥600 µmol/L, and sapropterin will only benefit approximately 30–40% of patients with mild or moderate PKU [2,3,4]. Even with sapropterin, patients will rarely attain an unrestricted diet [5]. Therefore, the main treatment is a lifelong Phe-restricted diet supplemented with Phe-free amino acids or low Phe casein glycomacropeptide, usually with added vitamins and minerals [6]. Foods high in protein such as meat, fish, eggs, cheese, nuts, seeds, regular bread, cereals, pasta and flours must be avoided in patients with classical PKU.

An important part of a Phe-restricted diet is the provision of naturally low-protein foods (protein < 0.5 g/100 g or Phe < 25 mg/100 g), mostly vegetables, fruit, fats and sugars. These are accompanied by specially modified low-protein foods, e.g., low-protein bread, pasta, flours, milk replacements, cereals, pizza bases, burger and sausage mixes, biscuits, crackers, and cakes. These foods help to avoid catabolism (and subsequent elevation of blood Phe) through the provision of energy, they aid satiety, and they improve the variety of foods available in an otherwise monotonous and limited diet [7]. There are approximately 150 individual low-protein prescription foods (LPPF) available in the UK [8]. In Italy, Germany and the United Kingdom (UK), the percentage of mean daily energy intake from LPPF is reported as 47, 39 and 33% respectively [9,10,11]. The range and amount consumed varies depending on individual preferences, product availability, and PKU severity, with patients with milder phenotypes being less reliant on LPPF due to a higher natural protein tolerance.

In the UK, LPPF are classified as Foods for Special Medical Purpose (FSMP) and are approved by the government Advisory Committee on Borderline Substances (ACBS) based on clinical need, their efficacy, and price [12]. They are not available in retail shops or pharmacies and are usually obtained via National Health Service (NHS) prescriptions through the General Practitioner (GP). In England, there are 2359 patients with PKU (923 paediatric and 1436 adult patients) who need access to LPPFs [12]. Prescriptions for LPPF are issued monthly and dispensed by specialist home delivery companies or local pharmacists. In Scotland, Wales and Northern Ireland (NI), all prescriptions for all patient age groups are issued free of charge through the NHS and Health and Social Care (NI). In England, prescription products are without cost to children under the age of 16 years or those in fulltime education aged ≤18 years, but most adults (aged ≥19 y to 60 y) are expected to pay a prescription fee for each LPPF item requested unless they are exempt or they have purchased a prescription prepayment certificate at a subsidised rate [13]. In order to control expenditure, guidance on the maximum number of units (1 unit = 1 pack/box as sold) of any particular product to prescribe each month are available through the National Society for PKU (NSPKU) for different age groups [14]. See Figure 1 for flowchart of LPPF ordering and acquisition process.

In PKU, evidence from studies in Scotland and England [7,12] suggest that the majority of LPPF prescribed by the NHS are staple foods such as bread, pasta, flour mixes and cereals, with sweet foods such as cakes, biscuits and desserts used in smaller amounts. Numerous studies have acknowledged significant issues with the current system for distribution of LPPF in the UK [7,12,15]. These include but are not limited to difficulty in patients obtaining GP prescriptions, incorrect prescriptions, incorrect product quantity received, delayed delivery of products, out-of-stock products, GPs declining requests or restricting prescription amounts, out-of-date, damaged or poor-quality products. Of the 252 patients with PKU or their caregivers completing an online questionnaire, 59% reported problems with accessing basic LPPF, such as bread and pasta, and over a third had recurring problems [15]. Patients/caregivers reported stress or anxiety associated with the process of obtaining LPPF [7,15], and there was an impression that PKU was not taken seriously by GP practices (clerks, managers, pharmacists and GPs), prescribers and suppliers. In addition, over 50% of patients/caregivers felt that this affected their health or the health of their child [15].

It is well-established that LPPF are more expensive than their equivalent regular or ‘free from’ products [12], leading to a GP’s reluctance to prescribe them. In addition, prescription charges in England may discourage patient use of LPPF, potentially causing termination of dietary treatment [12]. In children who receive free LPPF, the amounts prescribed per patient are less than 50% of the maximum amounts recommended by the NSPKU [7,12], and the cost to the NHS (<GBP 1000/patient/year) [12] is well below previously estimated costs (GBP 2000–GBP 2500/patient/year) [15]. The underutilisation of LPPF may well be associated with the difficulties inherent in the prescribing and dispensing system.

Whilst issues with the current system for prescribing and dispensing LPPF have been reported from the patient perspective [7,12,15], any challenges from a dietetic perspective have not been previously described. Problems with deliveries of LPPF are generally referred to the patient’s dietitian and/or dietetic support workers (DSW). This survey aimed to investigate the extent of any problems experienced by UK dietitians, DSWs and dietetic administrators in organising LPPF for their patients and correcting errors in their prescription, dispensing and delivery.

## 2. Materials and Methods

### 2.1. Study Group

In February/March 2024, an online questionnaire via Microsoft Forms (Microsoft 365) was emailed to all British Inherited Metabolic Disease Group (BIMDG) dietitians (*n* = 100) and cascaded to DSWs and administrative staff, inviting individuals involved in managing LPPF ordering for PKU to respond.

### 2.2. Questionnaire

The non-validated questionnaire (see Appendix A) was designed and piloted with a focus group of 4 dietitians acting in an advisory capacity for the NSPKU and 20 BIMDG dietitians and was further modified before distribution. The questionnaire comprised nine demographic multiple choice/short-answer questions identifying workplace, UK location, role (clinical dietitian, DSW, administrator), days worked per week, specialising in adult or paediatric care (or both), dietetic prescribers, and any patient prescription charges. The questionnaire was divided into paediatric and adult sections and respondents were directed to each section based on which patient group they work with. Each section comprised 27 multiple choice questions and 21 extended response questions inviting the respondent to comment on their previous answers. A summary of the questionnaire content is given in Table 1.

### 2.3. Data Analysis and Statistics

Responses were analysed, and the quantitative data were presented as a percentage of responses and analysed statistically using nonparametric Fisher’s exact test (statistical significance *p* < 0.05, 95% confidence) to compare differences between paediatric and adult dietetic responses. Qualitative data from open comments was grouped into themes and tabulated. Statistical analysis was carried out using GraphPad Prism version 10.2.2 for Windows, GraphPad Software, Boston, MA, USA.

### 2.4. Ethical Approval

Ethical approval was not required as there was no patient involvement, no change in patient treatment, and no collection of respondent personal or identifiable data. Ethical review and approval were not required according to the NHS Health Research Authority and Medical Research Council “Is my study research” online decision tool [16] (Appendix A).

## 3. Results

### 3.1. Study Group

In the UK, 72 inherited metabolic disease (IMD) dietitians and 25 DSWs/administrators were identified by the BIMDG as working with patients with PKU. Participants responding to this survey consisted of 53/72 (74%) clinical dietitians and 18/25 (72%) DSWs/administrators from 24 hospitals across the UK. All areas of the UK were represented. See Appendix A for details of respondents. Some respondents cared for both adult and paediatric patients with PKU (*n* = 3 dietitians and *n* = 6 DSWs), so they completed both the paediatric and the adult questionnaire. A total of 80 questionnaires were completed (44 paediatric and 36 adult). All hospitals had a minimum of 10 patients with PKU, and 82% (*n* = 58) cared for >50 patients with PKU. Over half (52%, 37/71) of the respondents worked full-time (*n* = 28 dietitians and n = 9 DSWs).

### 3.2. Prescribing LPPF

Although six dietitians (11%) were dietetic prescribers (i.e., NHS approved supplementary prescribers), all respondents said that their patients’ LPPFs were prescribed by the patient’s GP (except patients in the area of Rotherham, Yorkshire where an independent pharmacist issued prescriptions). Sixty-three percent (45/71) of respondents said that all of their patients were exempt from prescription charges. This included all NI, Scottish and Welsh centres (*n* = 14) and English paediatric centres (*n* = 31). However, patients aged > 16 y in Scotland, Wales and NI paid a prescription charge if their LPPF was dispensed and supplied via an English home delivery/dispensing company.

For patients in England, dietitians were concerned about the transition from paediatric to adult care as prescription charges were considered a deterrent to lifetime dietary treatment.

### 3.3. How LPPF Are Dispensed

The majority of respondents (84%, 67/80) reported that LPPF were supplied to patients via a combination of home delivery companies and their local pharmacy. This was determined by the patients or their caregivers.

### 3.4. Dietetic Time Managing LPPF Supply

Fifty percent (40/80) of respondents received communication from patients/caregivers at least weekly about problems accessing LPPF, and at least 20% (*n* = 8) said they were contacted daily or several times a week.

Twenty-nine percent (23/80) of respondents spent 3 or more hours/week organising supplies of LPPF for their patients (Table 2). Paediatric dietitians/DSWs spent more time than adult dietitians/DSWs co-ordinating LPPF supplies although this was not statistically significant (Table 2). Similarly, 24% (19/80) spent 3 or more hours/week correcting problems with LPPF for their patients with more time expended on this task by paediatric than adult dietitians/DSWs (Table 2). This could cost the NHS up to GBP 6500 annually per respondent in lost dietetic resources.

### 3.5. Main Issues with LPPF Supply

The most common issues for which paediatric and adult patients sought help from their dietitian/DSWs were unavailable LPPF (81%, *n* = 65/80), missing delivery items (79%, 63/80) and delayed LPPF deliveries (commonly without communication) (66%, 53/80). Paediatric patients were more likely than adult patients to have missing, damaged or inedible (e.g., mouldy bread or broken crackers), or out-of-date LPPF, whilst adults were more likely than children to be without LPPF supplies, experience out-of-stock items, and require information about the process for re-ordering LPPF. However, these differences were not statistically significant.

Thirty-one percent (25/80) of respondents reported that up to 25% of their patients had recurring LPPF supply problems and 33% (26/80) said up to 50% of patients had repeated problems. There was no significant difference between paediatric and adult patients. Respondents reported that the most common items in which patients experience supply issues were staple foods, including low-protein bread (96%, 77/80), pizza bases (34%, 27/80), pasta and rice (41%, 33/80), milk replacements (35%, 28/80) and flour or bread mix (33%, 26/80) (Figure 2). This was similar for paediatric and adult patients, where no differences were statistically significant.

The top five most common reasons reported for LPPF supply issues were: GPs prescribing incorrect items (65%, 52/80), prescriptions received by the wrong dispenser (home delivery company or pharmacy) (60%, 48/80), pharmacists unable to locate a LPPF on their online database that list all the ACBS LPPF (58%, 46/80), manufacturer supply issues (54%, 43/80), or the GP’s were unable to locate LPPFs on their online drug database (53%, 42/80). There was no significant difference between paediatric and adult patients in terms of the reasons for supply issues.

As a consequence of supply problems, 38% (30/80) of respondents provided patients with some additional “bridging stock” of LPPF from their hospital’s own supplies at least monthly and 23% (7/30) at least weekly. This was significantly more problematic for paediatric patients. Fifty-two percent (23/44) issued LPPF to children monthly compared with 19% (7/36) to adults with PKU (*p* = 0.003; Fisher’s exact test).

### 3.6. Frequency of Issues with LPPF

Sixty percent (48/80) of respondents were informed about manufacturer delays in the supply of LPPF at least monthly; of these, 13% (6/48) were informed of problems at least weekly (Table 3). There were no differences between paediatric and adult respondents.

GP prescription issues (see Section 3.7 for main issues) occurred at least weekly for 38% (30/80) of respondents, and this was more common in adult patients (47%, 17/36) compared with paediatric patients (30%, 13/44) (Table 3). Nearly half of the respondents (48%, 38/80) had to resolve issues with GP refusal to send prescriptions directly to delivery companies at least monthly, with one quarter (26%, 10/38) occurring at least weekly. This was similar for paediatric and adult patients.

Respondents experienced issues with the pharmacy dispensing of LPPF at least monthly (98%, 78/80) and at least weekly for 25% (20/80) of the respondents (Table 3). There was no significant difference between paediatric and adult patients.

Eighty-one percent (65/80) of respondents reported delivery company issues with dispensing of LPPF at least monthly, and 28% (22/80) at least weekly (Table 3). This was similar for paediatric and adult patients, with no statistically significant differences for any of the contact issues.

### 3.7. Main Issues with GP Practice

The top five issues with GP prescription of LPPFs reported to dietetics were prescription delays (79%, 63/80), prescriptions being sent to the incorrect supplier (65%, 52/80), prescription items missing (50%, 40/80), the GP practice unable to locate a specific LPPF on their computer drug database (41%, 33/80), and the number of LPPF items was less than requested (40%, 32/80) (Figure 3). Issues for paediatric and adult patients were generally similar, although adult patients were twice as likely as paediatric patients to experience GP refusal of some LPPF items (50%, 18/36 vs. 25%, 11/44; *p* = 0.03 Fisher’s exact test) (Figure 3).

Nearly 50% (38/80) of respondents said GP practices put some restrictions on the prescription of LPPFs (50%, 18/36 adults, 45%, 20/44 paediatric patients). In addition, half of the respondents (50%, 40/80) reported that GP prescriptions were commonly electronically downloaded by the wrong supplier (home delivery company or pharmacy) using the NHS digital spine platform at least monthly and 25% (10/40) at least weekly.

### 3.8. Main Issues with Pharmacies

The top five problems reported when pharmacies dispensed LPPFs were a lack of manufacturer supply (69%, 55/80), inability to source some of the LPPFs (50%, 40/80), lack of pharmacist knowledge about LPPF (45% (36/80), missing delivery items (44%, 35/80), and issues accessing the GP prescription (43%, 34/80) (Figure 4). One concern that was more prevalent for adults compared with paediatric patients, was lack of manufacturer supply (83%, 30/36 vs. 57%, 25/44; *p* = 0.01 Fisher’s exact test) (Figure 4). Conversely, poor communication with caregiver/patient when products were unavailable was more problematic for paediatric than adult patients (39%, 17/44 vs. 17%, 6/36; *p* = 0.05) (Figure 4).

### 3.9. Main Issues with Home Delivery Companies

All respondents (100%, *n* = 80) reported that most patients used at least one or more home delivery companies for deliveries of LPPF.

Twenty-nine percent (23/80) of respondents said that home delivery companies contacted them at least weekly for help in obtaining prescriptions from GPs and 79% (63/80) at least monthly.

The top five problems reported with home delivery companies for patients accessing LPPF were home delivery company difficulty obtaining the prescription from the GP (65%, 52/80), missing items in the delivery (60%, 48/80), poor communication between the GP and the home delivery company (59%, 47/80), inadequate communication with the patient or caregiver about delivery dates/times (44%, 35/80), and delayed deliveries (43%, 34/80) (Figure 5). Issues were similar for paediatric and adult patients, although adults were more likely than children to experience problems with home delivery companies accessing the prescription from the GP (78%, 28/36 vs. 55%, 24/44; *p* = 0.04), and locating the prescription on the NHS electronic spine (22%, 8/36 vs. 5%, 2/44; *p* = 0.04 Fisher’s exact test) (Figure 5).

### 3.10. Patient/Caregiver Issues

The most common patient and caregiver issues influencing the supply of LPPF were late ordering of LPPF (81%, 65/80), not responding to messages about the delivery (73%, 58/80), patient and caregiver confusion about the ordering process (70%, 56/80), and 28% (10/36) of respondents said that adults with PKU could not afford to pay for LPPF prescriptions. Respondents reported a trend for adults with PKU to have more problems with understanding the ordering process than caregivers of children with PKU (81%, 29/36 vs. 61%, 27/44; *p* = 0.08 Fisher’s exact test).

Thirty-nine percent (31/80) of respondents reported that prescription issues with LPPF frequently impacted metabolic control. There was no significant difference between adult and paediatric respondents. Figure 6 summarises the interdependent issues in the acquisition of LPPF for PKU that can adversely affect metabolic control.

### 3.11. Summary of Issues with the Current System for Obtaining LPPF in the UK

Table 4 summarises the open comments from dietitians and DSWs divided into common themes with supporting quotes.

### 3.12. Improvements Required

The most commonly reported recommendations by respondents for improving access to LPPF for patients with PKU included centralisation of the system to one supplier (76%, 61/80), and apps for ordering LPPF (69%, 55/80) (Figure 7). Paediatric respondents were more likely to suggest that improved GP surgery education may help (80%, 35/44 vs. 56%, 20/36 for adult dietitians/DSWs, *p* = 0.03 Fisher’s exact test), and adult dietitians/DSWs were more likely to suggest direct dietetic prescription of LPPFs (67%, 24/36 vs. 48%, 21/44 paediatric dietitians/DSWs; *p* = 0.11) (Figure 7).

Based on open comments in the questionnaire, Table 5 shows a list of possible solutions or improvements to the current system.

## 4. Discussion

The results of this survey showed that organising and managing patient access to LPPF for PKU was time-intensive and arduous for dietitians/DSWs working in paediatric and adult services and challenging for patients. Fifty percent of dietitians/DSWs were notified at least weekly that some of their patients had problems accessing LPPFs. Common issues with LPPFs were unavailability, missing items in deliveries, delayed deliveries, and prescription refusal by GPs. A central NHS digital prescription service, not designed to manage access to LPPFs for rare IMD conditions, generated some of the supply difficulties, coupled with increasing issues in the availability of some LPPFs.

There was unequal access to ‘free’ prescriptions for patients across the UK. Although there are only 1436 adults with PKU in follow up in English hospitals [12], unlike patients in Scotland, Wales or NI, they had to pay for the cost of their prescriptions. The cost of an NHS prescription in 2024–2025 was GBP 9.90 per item. Alternatively, an adult could purchase a certificate for prepayment for prescriptions (PPCs), in which, for a preset price of GBP 32.05 once every three months, or GBP 114.50 every twelve months, all prescriptions are covered, including LPPFs and protein substitutes [13]. This creates a substantial cost burden for patients with PKU and produces a barrier to treatment access. Inadequate access to LPPF may force patients to either abandon their treatment or go hungry. It has been previously suggested that prescription charges for working-age people may have a significant impact on adherence, self-management, quality of life and health outcomes [17].

Low-income or benefit-qualified patients may be eligible for help with medication costs. People entitled to free prescriptions in England include individuals aged 60 or over, aged under 16 years, aged 16 to 18 years and in full-time education, and pregnant or having had a baby in the previous 12 months. However, this can lead to confusion for patients, and some may be unaware of the need to pay for prescriptions. Some patients with PKU have mistakenly claimed free prescriptions (believing they were exempt) and have received financial penalty charges [18]. Furthermore, patients also need strong literacy and numeracy skills to be able to apply for some of the exemptions. They need to supply dated certificates to ‘prove’ exemption and reapply regularly for their exemption documents. It is possible that the cost to the NHS of providing exemptions to working-age people with PKU would be more than offset by reductions in healthcare resource use [17]. This would not only be through the cost savings associated with a reduction in administration and dietetic time but would enable patients to sustain satisfactory and stable metabolic control, potentially leading to fewer co-morbidities and lower demands on the health care system overall.

In addition, dietitians describe how some patients with PKU have difficulty accessing LPPFs on prescription, leaving them without adequate supplies. In this survey, respondents reported that some GPs did not prescribe sufficient quantities of LPPF or home delivery companies had issues accessing prescriptions for the LPPF from the GP practice, particularly in adulthood. Adult patients reluctantly tolerate these issues even though they may be left without LPPF. This leads to abandonment or relaxation of dietary treatment and the consumption of higher protein replacement foods purchased from supermarkets. Dietitians describe how some GP practices do not accept the seemingly high amounts of LPPFs that adult patients may need for a 28-day supply, leading to under prescription particularly of staple foods like bread. Patients/caregivers may also underestimate the amount of LPPF they need for a full four-week prescription cycle. Forecasting accurate amounts of necessary supplies is difficult, especially if there is more than one child in the family with PKU or LPPFs arrive damaged or mouldy and are inedible. Some patients/caregivers have difficulty understanding the ordering process and prescription timelines. Dietitians describe how some patients/caregivers are confused, frustrated and overwhelmed by the over complicated ordering systems leading to unnecessary stress and anxiety. Prescription problems commonly re-occur for the same patients and this can be demotivating for patients/caregivers, and may lead to food poverty and loss of metabolic control. Inadequate supply of LPPF may be a particular problem in maternal PKU where strict control of blood Phe is crucial for healthy foetal outcome.

A large amount of dietetic time was spent on unnecessary tracking of missing prescriptions for LPPF supplies. There were often delays in GPs issuing prescriptions for LPPFs. This was associated with prescriptions being issued to the wrong pharmacy/home delivery company, the wrong quantity of prescription items being given, some LPPF items being omitted, and mistakenly categorising LPPF as gluten-free and then refusing to issue a new prescription. Liaising with GP practices may be frustrating, with long telephone queues or untimely delays in responding to emails. Dietitians cannot access the GP systems to check if prescriptions have been completed, if all the items have been issued correctly or where the prescription has been sent. In addition, home delivery companies frequently request dietitians to obtain prescriptions from the GP on their behalf due to frequent unsuccessful communications with a GP practice. It is recognized that GP practices are exceptionally busy and overextended, and safe administration of multiple item prescriptions for LPPFs is likely to be time-consuming. Many will have received little training on LPPFs, the amount that should be prescribed, and they may have difficulty identifying specific LPPF products on their computer systems.

Home delivery services for LPPFs have seen considerable expansion in recent years. Most respondents reported that their patients accessed their LPPFs by using one or more home delivery companies in combination with their local pharmacy. In theory, home delivery services should enable patients to receive their prescription items in a consistent, safe and timely fashion, but the NHS system for issuing prescriptions does not readily support the use of multiple home delivery companies to provide a smooth service to patients. In England, with the electronic prescription service (EPS), patients can nominate for their prescriptions to be sent electronically and directly from the GP surgery to one single pharmacy or dispenser of their choice. Nominations can be changed or removed, but only at the request of the patient. For any prescriptions without an EPS nomination, a token is issued containing a unique barcode which is uploaded to the NHS Spine. This can be downloaded, and the prescription details retrieved by any home delivery company or pharmacy. 

In our survey, dietitians reported numerous problems with this service with prescriptions commonly being lost or downloaded by the incorrect delivery service, leading to frustration, stress and immeasurable loss of professional time tracking prescriptions. Better collaboration and understanding is necessary between the NHS digital and management teams, health professionals in primary and secondary care (including GP practice clerks, managers, pharmacists and GPs), and supply chain partners, including manufacturers, suppliers, distributors and logistics providers.

Patients can register to use the NHS App to request repeat prescriptions. Using this system, patients will know when their prescription is issued and can see all their prescription details in one secure place, giving them greater visibility and control of their healthcare. Each repeat prescription ordered electronically, saves GP practices three minutes of time [19]. However, this system does not work for non-repeat prescriptions. This is unfortunate, as the majority of LPPF prescriptions required by patients with PKU are non-repeat. With over 150 LPPFs on prescription, the amount and type of each LPPF ordered is likely to vary from month to month, meaning that non-repeat prescriptions must be ordered by directly contacting the GP surgery each month. In addition, patients with poor digital literacy, without appropriate technology or with learning disabilities may be disadvantaged by digital prescription ordering.

In the UK, each prescription is written by a registered prescriber, i.e., GP. Traditionally, the prescribing of human medicines has been perceived as a medical role, with only medical professionals and dentists having full prescribing rights. Two seminal reports challenged this view; the Cumberlege Report [20], which paved the way for limited prescribing by health visitors and district nurses, and the Crown Report [21], which recommended extending prescribing rights for the benefit of patients and to utilise the skills and extend the role of healthcare professionals. From February 2016, dietitians were able to become supplementary prescribers following the successful completion of a Health and Care Professions Council (HCPC)-approved prescribing programme [22]. However, only six dietitians in this survey were qualified prescribers. In addition, these dietitians did not have a financial budget to support the prescription of LPPF. Without the financial resources, it is unmanageable to further develop dietetic prescription services.

Some home delivery services did not meet the needs of patients with PKU. Commonly, delivery times were inflexible with a weekday service only, meaning that working adults or caregivers were not at home to accept deliveries. At times, supplies from the same company arrived in separate deliveries on different days, causing confusion for patients and caregivers. Adult patients living in Wales and Scotland (normally exempt from prescription charges) were expected to purchase a prescription certificate of pre-payment in order to access services from English home delivery services. Dietitians reported that most adult patients were reluctant to do this. They also report that patients were commonly unaware of expected delivery dates for their LPPF, and so contacted their specialty dietetic team for this information. Equally, there were issues with community pharmacists. Dietitians reported pharmacy delays in receiving LPPFs from multiple suppliers, dispensing of short shelf-life stock received from wholesalers, and patients/caregivers being informed that some LPPFs have been discontinued when there were stock issues only. Commonly, dietitians reported having to supply patients with ‘bridging stock’ or samples to ensure that they had sufficient LPPF to eat.

A frequent issue identified by dietitians was the interruptions to LPPF supply and manufacturing issues. Manufacturers of LPPF have significant challenges. They require alternative and suitable low-protein grains/starches requiring complex manufacturing processes. Since 2011, the UK NSPKU have identified that 43 LPPFs have been withdrawn by manufacturers and thereby from prescription [23]. Manufacturers may have fragile supply chain challenges, increasing energy costs, costs associated with inflation, delays in accessing key ingredients and raw materials, batch production failures, global supply chain challenges, and labour shortages. Goods imported may take longer to arrive in the country, and the introduction of full customs controls in the European Union has created additional administration. LPPF manufacturers supplying a range of products may have to use third-party suppliers who consider that it is not cost effective to produce goods in the small volumes required by patients with rare disorders such as PKU. Furthermore, outsourcing production means reliance on external partners to maintain the quality of LPPF.

For the future, dietitians considered that a single centralised, dedicated delivery company dispensing all LPPF would make the prescription service easier, although this would need service evaluation. In addition, the flexibility of patients and caregivers being permitted to directly request a set number of LPPF items each month, possibly through an app system monitored by their specialist dietetic team rather than the GP service, would improve the effectiveness and efficiency of LPPF access. A regulated age-related monthly financial allowance could be considered, giving patients the freedom to order the LPPF they need directly from the suppliers. Similar systems are implemented in countries such as Australia, Belgium, Denmark and Norway [24]. Although regular GP service education is important, staff turnover and the number of people involved make this unviable. As of November 2024, there are over 35,000 fully qualified GPs working in England alone, with less than 100 IMD dietitians to provide the education [25]. However, more educational resources, such as short videos, could be developed to assist patients with LPPF ordering.

### Limitations

Respondents had varying levels of involvement in managing patient LPPF access issues, and this will have influenced their answers, particularly with respect to the amount of dietetic time spent in organising LPPFs and correcting problems, and the frequency of specific problems. Some clinical dietitians, particularly in adult services, reported that these issues were more likely to be dealt with by their DSW than by themselves. Some hospitals did not have DSWs, and not all dietitians and DSWs worked exclusively in PKU or IMD. The number of years of experience working in PKU, and therefore the degree of exposure to the issues addressed by the questionnaire also varied. The average length of time to complete the questionnaire was 50 min (range: 6–198 min), which in turn may have led to respondent fatigue and accuracy of answers in later questions in the questionnaire. Some adults with PKU that may be less likely to report issues with LPPF supply compared with caregivers, masking the full extent of the problems observed.

## 5. Conclusions

Prescription of LPPF for patients with PKU in the UK is complex, inefficient, and involves multiple stakeholders, leading to suboptimal LPPF patient supplies. LPPF prescriptions are generally issued by GP practices who do not receive any training or guidance in this task. It is a cause of patient and caregiver anxiety, stress and a potential source of error leading to overall mistrust in the healthcare system. The current system requires a substantial amount of IMD dietetic resources to ensure that patients have adequate LPPF supplies. Easier and seamless channels are necessary to support communication between GPs, patients, caregivers and supporting health professionals. A review of patient access to LPPF is urgently required.

## Figures and Tables

**Figure 1 nutrients-17-00392-f001:**
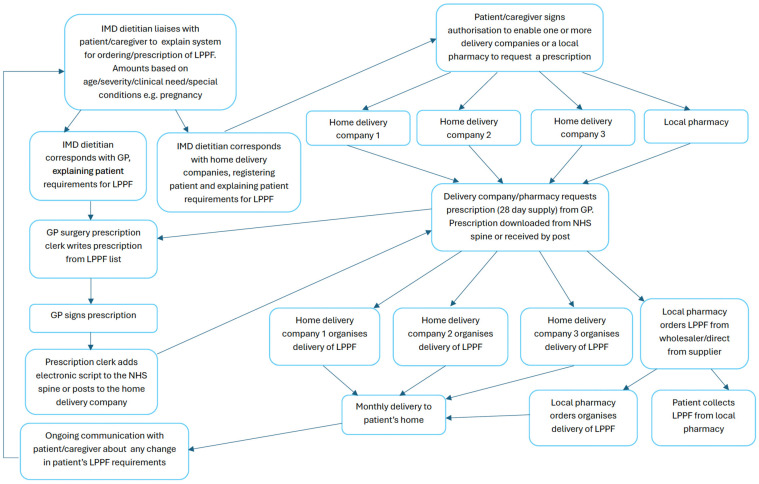
Low-protein prescription food ordering and acquisition process. IMD—Inherited Metabolic Disease; LPPF—Low-protein prescription food; GP—General Practitioner; NHS—National Health Service.

**Figure 2 nutrients-17-00392-f002:**
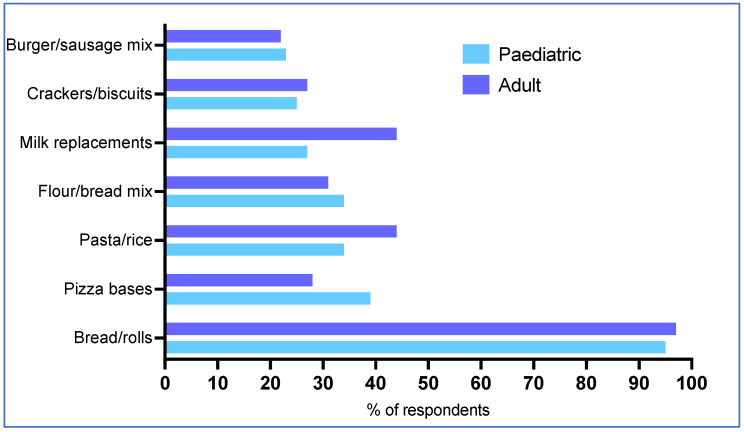
The most common LPPF supply issues patients experience.

**Figure 3 nutrients-17-00392-f003:**
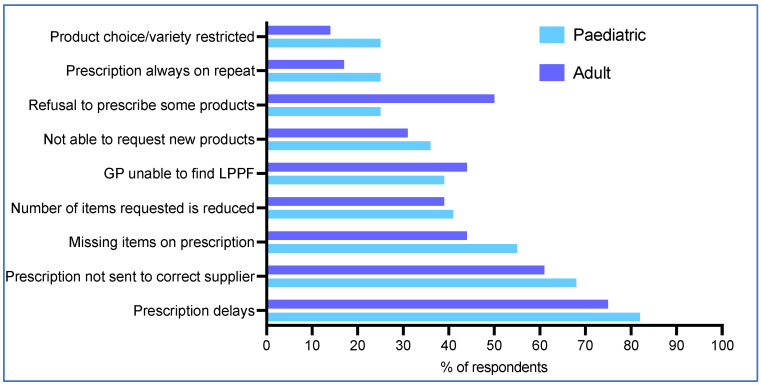
Main problems reported by caregivers/patients with GPs when obtaining LPPF.

**Figure 4 nutrients-17-00392-f004:**
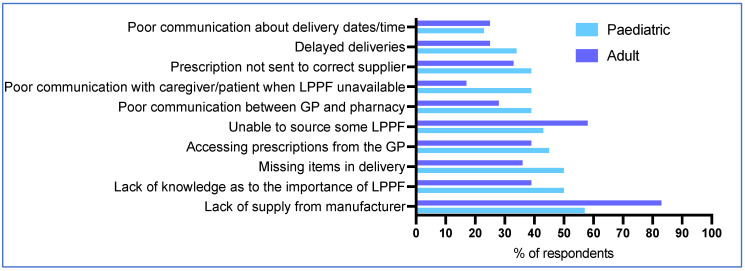
Main problems reported by caregivers/patients with pharmacies when obtaining LPPF.

**Figure 5 nutrients-17-00392-f005:**
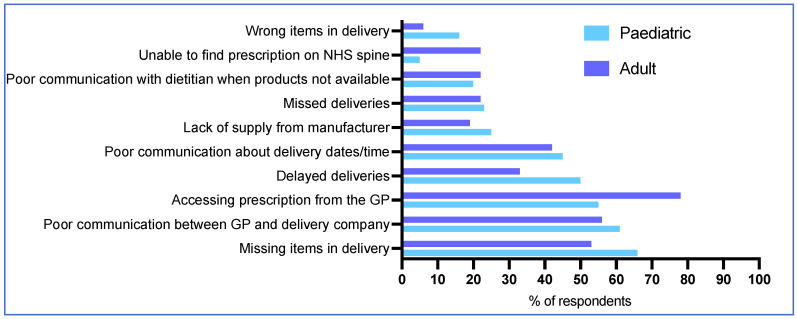
Main problems reported by caregivers/patients with home delivery companies when obtaining LPPF.

**Figure 6 nutrients-17-00392-f006:**
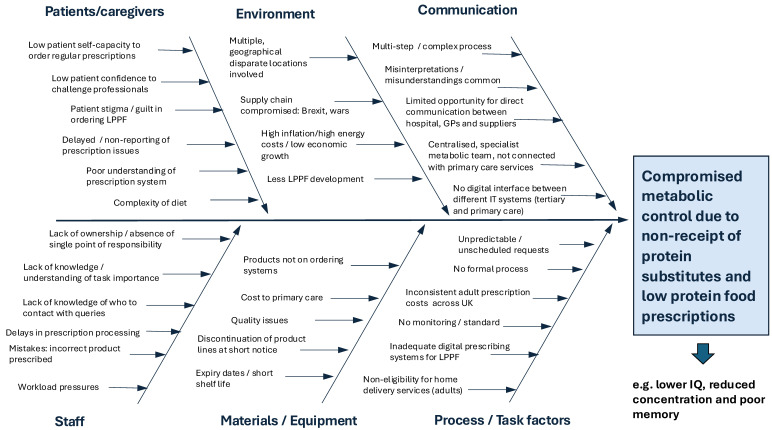
Interdependent issues associated with accessing LPPF for patients with PKU that may lead to ongoing supply issues and ultimately compromised metabolic control (LPPF—Low-protein prescription foods).

**Figure 7 nutrients-17-00392-f007:**
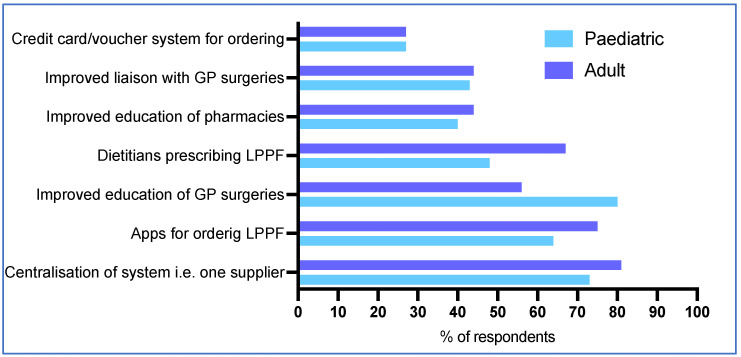
Suggestions for improving access to LPPF for patients with PKU in the future.

**Table 1 nutrients-17-00392-t001:** Topics addressed by the questionnaire regarding the supply of LPPF for patients with PKU.

Questionnaire Topics:
Number of patients with PKU in respondent’s dietetic care; Who prescribes and supplies LPPF for their patients; Frequency and nature of communication from patients regarding issues with accessing LPPF; Proportion of patients experiencing access issues to LPPF; Dietetic time spent in organising and correcting issues with LPPF supply; Type of LPPFs most difficult to access and why; Description of issues with LPPF manufacturers, GP prescriptions, pharmacies and home delivery companies; Frequency of communication with home delivery companies regarding patient supply of LPPFs; Frequency of GP LPPF refusal or any restrictions on supply; Issues with the NHS spine (the digital platform that connects healthcare information technology systems to enable secure sharing of prescriptions with pharmacy suppliers); Frequency of the need to issue emergency supplies of LPPF from the specialist PKU hospitals; Most common problems associated with patients/caregiver behaviour that affect the supply of LPPF; Impact of LPPF issues on patient blood Phe control; Suggestions for improving the current system for supply of LPPF.

LPPF—low-protein prescription food.

**Table 2 nutrients-17-00392-t002:** Dietetic time per week spent on organising LPPF and correcting LPPF problems n (%).

Hours/Week	Organising LPPF *n*(%)		*p* Value *	Correcting LPPF Problems *n* (%)		*p* Value *
	Paediatric	Adult	0.08	Paediatric	Adult	0.09
<1 h	13 (30)	13 (36)		12 (27)	19 (53)	
1–2 h	15 (34)	16 (44)		21 (48)	9 (25)	
3–5 h	12 (27)	2 (6)		7 (16)	4 (11)	
>5 h	4 (9)	5 (14)		4 (9)	4 (11)	

LPPF—Low-protein prescription food * Fisher’s exact test (paediatric vs. adult).

**Table 3 nutrients-17-00392-t003:** Frequency of patient dietetic contact regarding issues with LPPF supply *n*(%).

Contact Issue	At Least Weekly	Every 2 Weeks	Once a Month	Rarely	Unaware It Is an Issue
Manufacturer delays	6 (8)	11 (14)	31 (39)	31 (39)	1 (1)
GP prescription issues	30 (38)	25 (31)	21 (26)	4 (5)	0 (0)
GP prescription refusal	10 (13)	15 (19)	13(16)	24(43)	8 (10)
Pharmacy issues	20 (25)	25 (31)	33 (41)	2 (3)	0 (0)
Delivery company issues	22 (28)	24 (30)	19 (24)	14 (18)	0 (0)

**Table 4 nutrients-17-00392-t004:** Issues with the current system for obtaining low-protein prescription foods for PKU.

Issues	Supporting Respondent Quotes
**Prescription charges** In England, prescription charges are exempt for children <16 y or in full-time education ≤18 y. Adult patients in England have to pay a prescription charge for each low-protein food item or obtain a subsidised pre-payment certificate; In Scotland, Wales and NI, prescriptions are free for all patients; Some delivery companies have dispensing pharmacies in England, so patients in Scotland, Wales and NI have to pay a prescription charge for deliveries; NHS spine (electronic prescribing system) is not available in Scotland, Wales and NI so prescriptions are sent by post to home delivery companies.	*I worry for when patients become adults and how having to pay prescription charges may affect compliance to the very low-protein diet* [English response]. *Causes a lot of confusion for those turning 18* [English response]. *Prescriptions are free for all ages in Scotland. It does however cause issues with the home delivery companies who all have dispensing pharmacies based in England* [Scottish response]. *Adults in Wales are not eligible for some home delivery services, unless they pay a pre-payment prescription. As they get their prescriptions for free, most are reluctant to do this. LP foods predominantly supplied by local pharmacy* [Welsh response]. *In Northern Ireland, prescriptions do not go via ’the electronic spine’ and need to be sent via post—there are often issues with prescriptions going to the wrong place/getting lost [Northern Ireland response].*
**Dispensing of low-protein prescription foods** Different delivery companies deliver different LPPFs; Use of more than one delivery service is confusing for GPs, delivery companies, pharmacies and patients; Use of more than one delivery service increases the number of home deliveries each month, this is made worse by some companies splitting deliveries so they arrive at different times/days; Some GPs operate a single nominated pharmacy policy so patients can only choose one delivery service/pharmacy.	*It is difficult for families to understand the prescribing process and know who/when to contact if there are problems.* *It varies from patient to patient depending on how organised they are, how good their pharmacy is and the range of products the child accepts.* *Having different suppliers of low-protein prescription foods is confusing for families, GPs and pharmacists. It can lead to prescription requests being sent to the wrong pharmacy or home delivery company and this ultimately results in a delay in the time it takes for patients to receive their low-protein foods. This can then have a knock-on effect on compliance as these patients cannot just go and buy these foods from the local supermarket.* *Adult patients need to be at home to accept deliveries.* *Most GPs seem to operate a single nominated pharmacy policy and this causes so many problems getting the prescriptions to the right company/pharmacy, so patients can no longer have three different home delivery companies and get non-PKU related items from the local pharmacy.*
**Time dealing with low-protein prescription food issues** Dietitians suggest that problems are significantly under-reported; The reason why a patient cannot obtain their products is not always clear, so it requires a lot of phone calls, waiting on hold and waiting for responses; Most patients have issues at some point, and many have ongoing issues every month despite interventions; Much time is spent trying to access prescriptions from the GP because the delivery companies may refuse to call GPs.	*It is so time-consuming sorting out problems with low-protein prescription deliveries.* *Sometimes the prescription issue is resolved, then after a few months, the issue arises again.* *I feel for the patients and the families as it is time-consuming for them and a big additional burden to try and resolve their low-protein food prescription issues.* *A lot of dietetic time is spent ringing and waiting on hold at GP surgeries and being the go-between for communication with GPs, delivery companies and patients.*
**Pharmacy issues** Out-of-stock foods; Discontinued foods; Missing items from delivery; Delivery of short-shelf-life products; Patients informed that the low-protein food is discontinued when the wholesaler is out-of-stock, or it has been ordered from the wrong wholesaler, or pharmacies have not tried an alternative wholesaler; Pharmacies cannot locate low-protein foods in their prescription system.	*Often patients report: “pharmacy says that product is out-of-stock/product discontinued” when actually the pharmacy are trying to order from the wrong wholesaler or haven’t tried an alternative wholesaler.* *A high street pharmacy will say an item is out-of-stock or no longer manufactured, when in fact it’s that their suppliers don’t stock it and they are unwilling to order it in through other means.* *Often, items are missing from their order but no information is given about why items are missing.*
**Home delivery company issues** Lack of communication between the home delivery companies and GP regarding prescriptions; Dietitians/dietetic support workers receive daily communication from home delivery companies asking them to obtain outstanding GP prescriptions on their behalf; Delivery companies claim that GPs are more likely to listen to dietitians rather than themselves regarding prescriptions; Patients are often unaware of when their delivery is due; Deliveries are often delayed. Damaged products are commonly received by patients/caregivers due to courier services; Methods of ordering can be confusing for patients; There is a lot of paperwork involved in registering patients with home delivery companies, and every time there is a change in prescription to add or remove a food item or change the quantity.	*Dietitians receive a lot of emails from delivery companies asking them to liaise with GPs to organise owed prescriptions.* *I routinely hear of requests not being delivered by a home delivery company because they are waiting for a prescription.* *Have daily communications from home delivery companies stating they have not got the prescription from the GP, so they are unable to deliver.* *Parents need to be organised about when they should send in their order forms and keep track of when they should be expecting deliveries.* *If patients are unaware when the delivery is due and are not home, there is a risk of failed deliveries or damaged food if left somewhere inappropriate.*
**Electronic Prescribing System (EPS)/NHS Spine** Delivery companies and pharmacies may download the wrong prescription from the EPS (one intended for another company/pharmacy); Many dietitians do not understand how the EPS works.	*Would hope that if delivery companies download the wrong prescription that they will put it straight back on the spine as they are aware it’s not for them. This is not always the case with local pharmacies who may think it’s for them or want to dispense it.*
**GP issues** Incorrect amount of food on prescriptions due to misconceived perceptions of what is appropriate; Incorrect or missing foods on prescription; GP refusal to prescribe some foods, e.g., “luxury” or “unhealthy” items such as low-protein cakes, biscuits or chocolate; GP refusal to send prescriptions to third-party pharmacies (i.e., delivery companies), so instead paper prescriptions are given to patients to post to the delivery companies; Some GPs operate a single nominated pharmacy policy so patients can only choose one delivery service/pharmacy; Some GPs refuse to accept requests for repeat prescriptions from home delivery companies; Some GPs cannot locate LPPFs on their computer system in order to prescribe them; GP surgeries do not understand the complex ordering process.	*Our prescription letter has instructions for the GP on how to send the prescription to each of the home delivery companies, including via the post, spine and nominated pharmacy options. But still, they get this wrong.* *GPs sometimes refuse more ’luxury/convenience’ foods.* *GPs are sometimes suspicious the whole family are using the products to save money.* *Items that should be on repeat prescription (and often are listed as such on the GP system) seem to ‘drop off’ and then have to be re-requested.* *GPs often struggle to find individual LPPFs on their system.* *It is difficult to explain to surgeries that patients can order any food from a long list of low-protein food items and that the prescription required will change each month.* *GPs do not understand the process and are confused as to where the prescription needs to be sent.* *Some GPs have a mistrust of home delivery companies and seem to believe that the home delivery companies are trying to rip them off.* *As PKU is a rare disorder, educating GP surgery staff is a relentless task due to the number of staff and staff turnover.* *Some GPs confuse LPPF with gluten-free foods which are available in the supermarkets.*
**Contacting GP surgeries** The first contact is usually the receptionist, who does not have the authority to resolve issues; The surgery pharmacist or prescription clerk is the preferred contact by most dietitians, but they are commonly unavailable (e.g., part-time); If issues are long standing or not resolving, dietitians will request to speak to the GP or practice manager.	*GP reception staff often do not have the correct training and are not sure what we are asking about.* *Practice pharmacists often work part-time—so it can be a week until you can speak to them.* *It depends on the practice. The first contact point is always the receptionist. Can be difficult to get beyond this, but I try to get put through to the person involved in prescribing.* *Phone lines can be extremely busy, and not all practices have dedicated prescription lines.*
**Manufacturer/wholesaler issues** Poor communication with patients and dietitians when foods are discontinued or out-of-stock; Increasing numbers of LPPFs are being discontinued; Some LPPF foods may be unavailable for periods of time.	*Dietitians are not always made aware that there are supply issues until we are chasing up why an order has not been received.* *A lot of popular items have been discontinued or the supplier has been having issues obtaining the products. These items are often staples (e.g., bread products).*
*** Patient/caregiver issues** Some patients do not understand the ordering process; Patients may not order in a timely way; The volume of LPPF required for 1 month is high, particularly for families with more than one child; Adults may be less likely to discuss issues with delivery companies due to learning and cognitive difficulties; The system is commonly overwhelming and difficult for patients; Some patients run out of LPPFs.	*Families do not allow enough time when ordering low-protein foods—but they are expected to be very organised.* *For some of our vulnerable patients, we arrange a standing order of staple LPPF to ensure they always have a supply, however they may change their minds about food they enjoy* *Families with more than one child can find it hard to store a month’s supply of LPPF.* *More problems tend to be in families with communication difficulties, e.g., due to language, level of education or in areas of lower socio-economic status.* *Many adult patients have learning difficulties and become frustrated with the system.* *The current system is clunky and inefficient and causes unacceptable stress and anxiety for patients. Many find it hard to engage with their GPs / pharmacies and they feel like they’re being a nuisance having to go back and ask for something else. They often give up.* *Patients may come to clinic and report they haven’t had any low-protein foods for the last 3 months.* *Accessing low-protein foods is a barrier to adults returning or staying on diet.* *Patients are frequently left without staple foods such as bread due to:* - *Missing a full delivery;* - *Missing products from their delivery;* - *Delayed deliveries;* - *Failure of GP to issue a prescription;* - *Damaged, mouldy or out-of-date food.*
**Impact on blood Phe** Inadequate supplies of LPPF can lead to high blood Phe levels due to: - Eating the wrong foods (higher in protein, e.g., gluten-free foods) as they have nothing else to eat; - Insufficient energy intake and weight loss leading to catabolism; - Limited variety of foods, which impacts enjoyment of food and subsequently dietary adherence; - Adult demotivation to follow dietary treatment; - Increased stress and anxiety in adult patients.	*Patients running out of LPPFs can lead to high Phe levels as patients have no choice but to eat the wrong foods. It also leads to hungry patients and weight loss.* *If bread arrives mouldy or not at all, they may have to use supermarket gluten-free options, which are higher in protein.* *Adult patients can become demotivated by poor product access.* *Often impacts blood Phe levels as adult patients can go on/off diet depending on what stock they have available.* *Some patients make do and only tell you when blood Phe levels are high that they have not been getting basic low-protein foods.*

* Note: The term “GP” is used interchangeably to refer to staff at GP surgeries who may be dealing with patient prescriptions (may be prescription clerk, pharmacist, receptionist, practice manager, GP). The term “patient” is used interchangeably to refer to adult patients or caregivers of children with PKU.

**Table 5 nutrients-17-00392-t005:** Possible solutions or improvements to the current system for accessing low-protein prescription foods for PKU.

Suggestions for Improvement
A separate UK prescribing system is necessary for LPPFs as they are different to drugs/medication; Centralisation of the system with one supplier/delivery company; Giving patients a budget each month to spend on LPPFs; Order LPPFs via a voucher or pre-loaded credit card; An app for ordering—one list, one supplier; Improve NHS App functionality for complex script handling; In the UK, all patients (of all age groups) with PKU should have access to free NHS prescriptions; Patients in Scotland, Wales and NI to have free access to specialist home delivery companies; Consider dietetic prescribing of LPPFs.

## Data Availability

The raw data supporting the conclusions of this article will be made available by the authors on request.

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
