# Peer review of "UK Patient Access to Low-Protein Prescription Foods in Phenylketonuria (PKU): An Uneasy Path"

_nutrients, 2025, doi:10.3390/nu17030392_

Round 1
Reviewer 1 Report
Comments and Suggestions for Authors
The manuscript presents a valuable study of collaborators from more than 20 institutions, devoted to the dietary treatment of phenylketonuria, based on the analysis of semi-structured questionnaires completed by British Inherited Metabolic Disease Group) dietitians and dietetic support workers (DSW)/administrators engaged in the treatment of this disease. The study identified main problems with the supply of low protein prescription foods identified by the responders.
The questionnaire is quite comprehensive and the analysis covers many problems related to the low protein prescription foods in the UK.
The manuscripts includes a section on suggested improvements. Limitations of the study are discussed, and brief Conclusion section is provide.
Remarks:
How many patients with phenylketonuria are registered in the UK?
Are there any problems with the quality of low protein food and keeping required standards?
Figure 1 lacks indicators of flow (at least in my inspection copy).
Table 2. It is not clear what is statistically compared: pediatric vs adult patients?; “tes” should be “test” in the footnote.
Author Response
Reviewer comments
The manuscript presents a valuable study of collaborators from more than 20 institutions, devoted to the dietary treatment of phenylketonuria, based on the analysis of semi-structured questionnaires completed by British Inherited Metabolic Disease Group) dietitians and dietetic support workers (DSW)/administrators engaged in the treatment of this disease. The study identified main problems with the supply of low protein prescription foods identified by the responders.
The questionnaire is quite comprehensive and the analysis covers many problems related to the low protein prescription foods in the UK.
The manuscripts includes a section on suggested improvements. Limitations of the study are discussed, and brief Conclusion section is provide.
Thank you for your helpful comments and suggestions. Responses are listed below and all changes in the manuscript are marked in red.
Remarks:
How many patients with phenylketonuria are registered in the UK?
Unfortunately we do not have statistics on this for the whole of the UK. However we have included a sentence in the introduction (line 93-94) to indicate the estimated number in England (2359) based on a recent study.
Are there any problems with the quality of low protein food and keeping required standards?
Thank you. The only standards that the low protein food manufacturers have to maintain are the same for normal foods. However, we do see problems with patients receiving moldy bread, broken crackers or damaged packets. We have added this to the text (line 224-225).
Figure 1 lacks indicators of flow (at least in my inspection copy).
Apologies, the arrows on Figure 1 don’t seem to show up on some computers. We have saved the figure in a different format so that the arrows are visible.
Table 2. It is not clear what is statistically compared: pediatric vs adult patients?; “tes” should be “test” in the footnote.
Thank you, a “t” has been added to the footnote as suggested and “(paediatric vs adult)” added to the footnote to indicate what is being compared (line 219).
Thank you again for your comments.
Reviewer 2 Report
Comments and Suggestions for Authors
Dear authors,
Research to support further alternatives to reduce issues related to low-protein prescriptions to PKU patients is worthy of being published. Please, consider the following points before manuscript publication:
Figure 1 - Revise the diagram and transform that into a flowsheet by specifying the steps precisely (use arrows to indicate from where each step for ordering begins through where it ends).
L166 - Please, specify which non-parametric test was used.
L169 - for Windows, not 'Window'.
L170 - There is not need to include the website, once all program details were mentioned.
L177 - That would be great include the pdf generated as supplementary material and write a sentence about that after the website citation. Also, instead of writing the website as it is, it is recommended including that in the reference section.
L207 - Please, correct the parenthesis .
L191-349 - For the data expressed between parenthesis: please include the table/figure from where the information was taken.
Figure 6 - One suggestion to improve: Include the minor topics grouped into individual blocks instead of including multiple arrows.
Author Response
Reviewer comments
Dear authors,
Research to support further alternatives to reduce issues related to low-protein prescriptions to PKU patients is worthy of being published. Please, consider the following points before manuscript publication:
Thank you for your helpful comments and suggestions. Responses are listed below and all changes in the manuscript are marked in red.
Figure 1 - Revise the diagram and transform that into a flowsheet by specifying the steps precisely (use arrows to indicate from where each step for ordering begins through where it ends).
Apologies, the arrows on Figure 1 don’t seem to show up on some computers. We have saved the figure in a different format so that the arrows are visible.
L166 - Please, specify which non-parametric test was used.
The test (Fisher’s exact test) has now been added (line 167).
L169 - for Windows, not 'Window'.
An “s” has been added to “Window” as suggested (line 170).
L170 - There is not need to include the website, once all program details were mentioned.
The website link has been removed as suggested (line 171).
L177 - That would be great include the pdf generated as supplementary material and write a sentence about that after the website citation. Also, instead of writing the website as it is, it is recommended including that in the reference section.
A PDF of the NHS HRA decision tool verifying that the study does not require ethical approval has been added as supplementary material (S2) and a note added to the text to indicate this (line 177-178). The website has also been removed from the text and added to the references (reference 17, lines 615-616)
L207 - Please, correct the parenthesis .
The parentheses have been corrected (line 208)
L191-349 - For the data expressed between parenthesis: please include the table/figure from where the information was taken.
Not all of the data in the results section is mentioned in a table or figure, where it is, the appropriate table or figure has been added to the text after the relevant data (lines 212, 293, 295, 314, 352).
Figure 6 - One suggestion to improve: Include the minor topics grouped into individual blocks instead of including multiple arrows.
Thank you for your suggestion. This is a fishbone diagram where all of the topics are equally relevant and of equal significance. As such, if it is okay, we would prefer not to group them together but keep them as individual points.
Thank you again for all your comments and suggestions.